materials science/nanotechnology

biomaterials, hydroxyapatite, poly(lactic acid), anti-cancer, drug delivery system

**Authors for correspondence:**
Sungho Lee
e-mail: sungho.lee@aist.go.jp
Fukue Nagata
e-mail: f.nagata@aist.go.jp

This article has been edited by the Royal Society of Chemistry, including the commissioning, peer review process and editorial aspects up to the point of acceptance.

# Development of paclitaxel-loaded poly(lactic acid)/hydroxyapatite core–shell nanoparticles as a stimuli-responsive drug delivery system

Sungho Lee[1], Tatsuya Miyajima[1], Ayae Sugawara-Narutaki[2], Katsuya Kato[1] and Fukue Nagata[1]

[1]National Institute of Advanced Industrial Science and Technology (AIST), 2266-98 Anagahora, Shimoshidami, Moriyama-ku, Nagoya 463-8560, Japan
[2]Department of Energy Engineering, Graduate School of Engineering, Nagoya University, Furo-cho, Chikusa-ku, Nagoya 464-8603, Japan

SL, 0000-0002-4367-9573; TM, 0000-0003-3084-9112; AS-N, 0000-0003-3579-6689; KK, 0000-0001-8981-8359; FN, 0000-0002-5284-9626

Biodegradable nanoparticles have been well studied as biocompatible delivery systems. Nanoparticles of less than 200 nm in size can facilitate the passive targeting of drugs to tumour tissues and their accumulation therein via the enhanced permeability and retention (EPR) effect. Recent studies have focused on stimuli-responsive drug delivery systems (DDS) for improving the effectiveness of chemotherapy; for example, pH-sensitive DDS depend on the weakly acidic and neutral extracellular pH of tumour and normal tissues, respectively. In our previous work, core–shell nanoparticles composed of the biodegradable polymer poly(lactic acid) (PLA) and the widely used inorganic biomaterial hydroxyapatite (HAp, which exhibits pH sensitivity) were prepared using a surfactant-free method. These PLA/HAp core–shell nanoparticles could load 750 wt% of a hydrophobic model drug. In this work, the properties of the PLA/HAp core–shell nanoparticles loaded with the anti-cancer drug paclitaxel (PTX) were thoroughly investigated *in vitro*. Because the PTX-containing nanoparticles were approximately 80 nm in size, they can be expected to facilitate efficient drug delivery via the EPR effect. The core–shell nanoparticles were cytotoxic towards cancer cells (4T1).

This was due to the pH sensitivity of the HAp shell, which is stable in neutral conditions and dissolves in acidic conditions. The cytotoxic activity of the PTX-loaded nanoparticles was sustained for up to 48 h, which was suitable for tumour growth inhibition. These results suggest that the core–shell nanoparticles can be suitable drug carriers for various water-insoluble drugs.

## 1. Introduction

Biodegradable nanoparticles have received much attention for their abilities to function as biocompatible delivery systems for biomolecules such as proteins, peptides, nucleic acids and oligonucleotides and to allow sustained drug release [1–3]. In particular, researchers have been taking advantage of the so-called enhanced permeability and retention (EPR) effect, where particles of up to 200 nm in size are able to diffuse into and accumulate in lymph vessels of tumour tissues [3–5] owing to the leaky and defective tumour blood vessel structures that form as a result of rapid vascularization to serve fast-growing tumour tissue [5]. Recently, the focus had been placed on stimuli-responsive drug delivery systems (DDS), such as those that are pH-sensitive or photo-responsive, for improving the effectiveness of chemotherapy [5,6]. For example, the extracellular pH of tumour tissue is weakly acidic at 6.5, whereas that of normal tissue remains constant at 7.2–7.4 [5,6]. Additionally, nanocarriers can be incorporated into the tumour cells via endocytosis and use the numerous pH gradients in cells; for example, the pH values of endosomes, lysosomes and mitochondria are 5.5–6.0, 4.5–5.0 and 8.0, respectively [5]. Hence, the size of the nanoparticles and their pH sensitivity can be strategies for designing drug carriers for DDS.

Poly(lactic acid) (PLA), poly(glycolic acid) (PGA) and their copolymer PLGA are widely used in the construction of DDS, owing to their biodegradability in the body [1]. For PLA and PLGA particles prepared using the solvent evaporation method, the size of the particles decreases with the increase of a stabilizer, such as poly(vinyl alcohol) [7]. However, the preparation of particles with a size smaller than 200 nm requires a large amount of stabilizers, which are non-biodegradable and tend to remain in the resultant nanoparticles [7]. In our previous work, PLA/hydroxyapatite (HAp) core–shell nanoparticles were prepared using a surfactant-free method [8–10]. HAp, which is a mineral component of bone and teeth, is used widely as an inorganic biomaterial owing to its excellent biocompatibility and biodegradability [11]. Moreover, HAp-containing drug carriers have been reported to have pH-sensitive drug release profiles [12,13], since HAp is stable under neutral pH conditions and dissolves in acidic environments [11]. For a drug carrier to be efficient, it should have a high drug-loading capacity and sustained drug-releasing capability [3,5] in addition to biodegradability and responsiveness to stimuli. Our previous PLA/HAp core–shell nanoparticles loaded a model hydrophobic drug (phylloquinone) of approximately 750 wt% and exhibited sustained release of the drug at 0.8% $d^{-1}$ with a linear profile [9]. Hence, PLA/HAp core–shell nanoparticles can be excellent drug carrier candidates for DDS.

In this work, we carried out a fundamental investigation on the *in vitro* properties of PLA/HAp core–shell nanoparticles loaded with an anti-cancer agent. Paclitaxel (PTX) was chosen as the model drug for loading, as it is a widely used anti-cancer agent [14]. This drug is poorly soluble in water and requires dissolution in ethanol and polyoxyethylated castor oil for administration to patients, which can cause serious side effects [15]. Therefore, the core–shell nanoparticles are expected to carry PTX without the need for ethanol and polyoxyethylated castor oil. The structure of the PTX-loaded PLA/HAp core–shell nanoparticles was evaluated and their effect on the viability of murine breast cancer cells was examined *in vitro*.

## 2. Experimental methods

### 2.1. Preparation of the PTX-loaded PLA/HAp core–shell nanoparticles

PTX-loaded PLA/HAp core–shell nanoparticles were prepared with modified surfactant-free emulsification methods, as described in our previous reports [9,10]. The PLA, PTX, calcium (Ca) and phosphate (P) ion solutions used for fabrication of the core–shell nanoparticles were prepared as follows. PLA (Resomer R 202 H, $Mw = 10$–18 kDa, Sigma-Aldrich) and PTX (BLDpharm) were separately dissolved in acetone at the concentrations of 20 and 1 mg $ml^{-1}$, respectively. Ca and P ions solutions were prepared dissolving 0.20 M calcium acetate monohydrate ($Ca(CH_3COO)_2 \cdot H_2O$, Wako

Pure Chemical) and 0.12 M diammonium hydrogen phosphate ($(NH_4)_2HPO_4$, Wako Pure Chemical), respectively, in ultrapure water.

The PLA and PTX solutions were added to ultrapure water, followed by aqueous Ca ion solution added. Next, aqueous P ion solution was added dropwise under stirring at 25°C. The final concentrations of PLA, PTX, Ca ion and P ion were 0.1 mg ml$^{-1}$, $x$ wt% of PLA ($x = 0$–5), 2.0 and 1.2 mM, respectively. After 72 h of stirring at 25°C to remove the undesirable acetone, each mixture was filtered through a 40 µm mesh and then centrifuged (6000 r.p.m., 10 min). Thereafter, the supernatant was removed, and the pellet was washed with ultrapure water. The various nanoparticle assemblies were resuspended at a concentration of approximately 2 mg ml$^{-1}$ (denoted by PTX$x$@HAp, $x = 0$–5 wt% of PLA). Removed supernatant of PTX0.0@HAp was filtered using ultrafiltration (Vivaspin 20, Sartorius, 10 kDa MWCO). The filtered supernatant was measured using inductively coupled plasma optical emission spectrometer (IRIS Advantage, Nippon Jarrell-Ash) to confirm the Ca/P ratio of the particle.

## 2.2. Structural characterization of the PTX$x$@HAp nanoparticles

The morphology of the PTX$x$@HAp nanoparticles was observed by field emission scanning electron microscopy (SEM, S-4300, Hitachi). PTX$x$@HAp solutions were directly dropped on the sample stage and dried overnight, and then they were coated with an amorphous osmium layer (Neoc-Pro, Miewafosis). The diameters of the drug-loaded nanoparticles were measured using ImageJ software (NIH). PTX0.0@HAp solution was directly dropped on carbon grid and dried; subsequently, the sample was observed with transmission electron microscope (TEM, JEM-2100F, JEOL). The weight ratio of HAp to (PLA + PTX) in the carrier assembly was evaluated by thermogravimetric (TG) analysis (TG-DTA8122, Rigaku). The crystalline structure of PTX$x$@HAp was evaluated by X-ray diffraction (XRD) (CuK$\alpha$, SmartLab SE/B1, Rigaku). The bonding structure of PTX$x$@HAp was analysed by Fourier transform infrared (FT–IR) spectroscopy with attenuated total reflectance (FT/IR-4700, JASCO) and by Raman spectroscopy (785 nm, XploRA, HORIBA). Freeze-dried PTX$x$@HAp powders were used for the TG, XRD, and FT–IR and Raman spectroscopy measurements.

## 2.3. Cell cultures and cytotoxicity tests

Murine breast cancer cells (4T1 cell line, CRL-2539, ATCC) were used for testing the cytotoxic effects of PTX$x$@HAp on cancerous cells. Roswell Park Memorial Institute (RPMI) 1640 medium (Wako Pure Chemical) containing 10% fetal bovine serum (Invitrogen) was used to culture the 4T1 cells. The PTX$x$@HAp nanoparticles were first sterilized by autoclaving at 121°C for 20 min (LSX300, TOMY SEIKO) and then centrifuged (6000 r.p.m., 5 min). After removal of the supernatant, the pellet was resuspended in a culture medium at various concentrations (20–1000 µg ml$^{-1}$). 4T1 cells were seeded ($2 \times 10^4$ cells ml$^{-1}$ in 100 µl of culture medium; $n = 5$) into 96-well plates and cultivated for 24 h (37°C, 5% $CO_2$). Then, the medium was replaced with a medium containing either PTX$x$@HAp or PTX alone (at various concentrations) and further cultivated for 24 or 48 h. The PTX-containing media used for comparison in the cell viability test were prepared by diluting a 5 mg ml$^{-1}$ filter-sterilized PTX–ethanol solution to various concentrations (0.05–25 µg ml$^{-1}$).

For the test of 4T1 cell viability, the Cell Counting Kit-8 (CCK-8, Dojindo) was used. In brief, after culture of the 4T1 cells for a prescribed time, they were washed with the medium and replenished with 100 µl of RPMI-1640 medium (no phenol red, Wako Pure Chemical), following which 10 µl of CCK-8 reagent was added. After 2 h of incubation, 100 µl of the reacted solution was transferred to a new 96-well plate and the absorbance at 450 nm was recorded. The number of live cells, which was determined from a standard curve of cell number versus absorbance of the resulting medium, was used to calculate the cell viability as a percentage of the control.

## 2.4. Fluorescence imaging of the core–shell nanoparticles and cancer cells

For fluorescence imaging, fluorescein-loaded PLA/HAp core–shell (denoted by Flu@HAp) nanoparticles were prepared. Fluorescein (Tokyo Chemical Industry) was dissolved in acetone at 1 mg ml$^{-1}$. PLA and fluorescein solutions were added to ultrapure water, followed by Ca and P ions solutions were added, as the same process as PTX$x$@HAp. 4T1 cells were seeded ($1 \times 10^4$ cells ml$^{-1}$ in 500 µl of culture medium) on a cover glass ($\varphi$ 15 mm) in 24-well plate and cultivated for 24 h (37°C, 5% $CO_2$). Next, the medium was replaced with a medium containing 1000 µg ml$^{-1}$ Flu@HAp and further cultivated for 24 h. Then, the cells were fixed with 4% formaldehyde in phosphate-buffered saline (PBS) for 20 min. The cells on

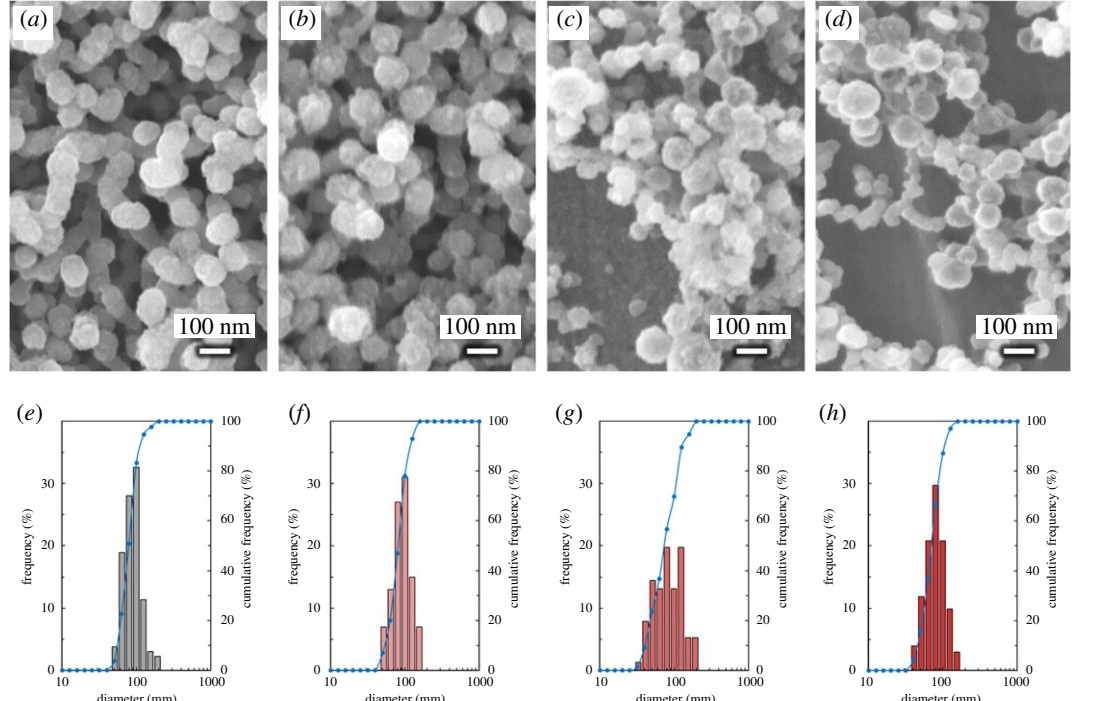

**Figure 1.** SEM images of (*a*) PTX0.0@HAp, (*b*) PTX1.0@HAp, (*c*) PTX2.5@HAp and (*d*) PTX5.0@HAp. Particle diameter distribution of (*e*) PTX0.0@HAp, (*f*) PTX1.0@HAp, (*g*) PTX2.5@HAp and (*h*) PTX5.0@HAp. Bar graphs represent the appearance frequency, and blue solid lines represent the cumulative frequency.

cover glass were washed three times with PBS-0.05% Triton X-100 (PBST), and incubated with Phalloidin-iFluor 594 Reagent (abcam) and DAPI (4′,6-diamidino-2-phenylindole, Wako Pure Chemical). Finally, the cells were washed with PBST and mounted in ProLong Gold (Invitrogen). Fluorescent images were taken using a fluorescence microscope (BZ-X800, Keyence).

## 3. Results and discussion

SEM images of PTX*x*@HAp are shown in figure 1*a–d*. The diameters of the nanoparticle assemblies are shown in table 1, and their size distribution is shown in figure 1*e–h*. TEM image of PTX0.0@HAp is shown in figure 2. Grey spherical parts, i.e. PLA core, were approximately 35 nm diameter, and black shell parts, i.e. HAp shell, were approximately 6 nm thickness. These sizes showed good agreement with our previous reports, where the PLA/HAp core–shell nanoparticles of 50 nm in diameter had a shell of 5 nm thickness [16]. PTX*x*@HAp core–shell nanoparticles were successfully prepared using a surfactant-free method, and their spherical shape was maintained despite the introduction of PTX. No other constructs, such as rod-shaped HAp and polymer aggregates, were observed. Moreover, the PTX*x*@HAp particle size was approximately the same or only slightly smaller with the introduction of PTX. The weight ratios of HAp : (PLA + PTX) for the various PTX*x*@HAp assemblies were all approximately 1 : 1 (table 1), similar to the results from our previous work [10]. This indicates that the addition of PTX up to 5 wt% of PLA does not induce morphological changes of the PTX*x*@HAp core–shell nanoparticles. It is known that nanoparticles of less than 200 nm in size exhibit the EPR effect in tumour tissue [3–5]. Given that the PTX*x*@HAp assemblies are in the much smaller size range of 75–85 nm, they are expected to facilitate passive targeting to tumour tissues and accumulation of the drug therein via the EPR effect.

The Ca/P ratio of PTX0.0@HAp was approximately 1.63, which was similar to the stoichiometric composition of hydroxyapatite with the value of 1.67. The XRD patterns of PTX*x*@HAp (figure 3) showed good agreement with those previously established for HAp (JCPDS database file no. 9-432) and also showed similar patterns with our previous PLA/HAp core–shell nanoparticles [9]. The XRD patterns of PTX*x*@HAp were not sharp, which are similar to biological hydroxyapatite [17], and indicated that HAp shell had low crystallinity. Additionally, the HAp shell formed on the end group of PLA [9], i.e. carboxyl group [18], and the shell composed polycrystalline HAp. The peaks of

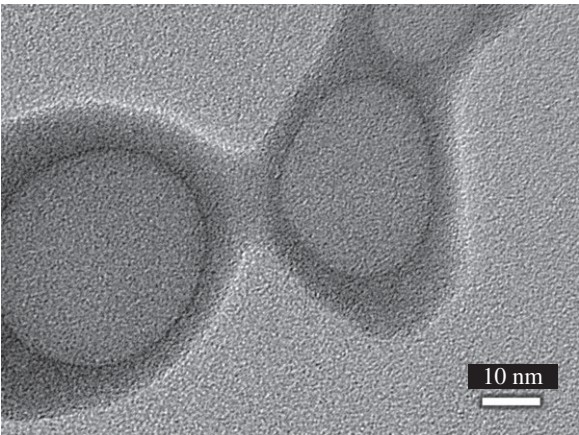

**Figure 2.** TEM image of PTX0.0@HAp.

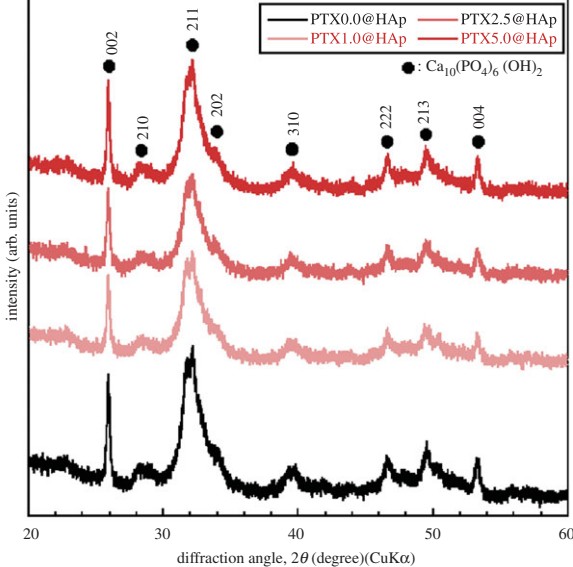

**Figure 3.** X-ray diffraction patterns of various PTX$x$@HAp assemblies.

**Table 1.** Particle diameters and HAp : (PLA + PTX) weight ratios of the various PTX$x$@HAp assemblies.

| SPL code | diameter (nm) | HAp : (PLA + PTX) |
|---|---|---|
| PTX0.0@HAp | $83.0 \pm 26.3$ | 53 : 47 |
| PTX1.0@HAp | $83.2 \pm 23.4$ | 54 : 46 |
| PTX2.5@HAp | $80.9 \pm 36.2$ | 51 : 49 |
| PTX5.0@HAp | $74.4 \pm 24.1$ | 52 : 48 |

PTX$x$@HAp showed no significant difference in the spectral patterns between nanoparticles with or without PTX. Remarkably, the peaks assigned to 002 and 004 were sharp and narrow in shape compared with the other peaks, which suggested that the crystalline $c$-axis arrangement of HAp in PTX$x$@HAp had a more ordered arrangement than that of the other axes. In our previous work, PLA/HAp core–shell nanoparticles of 50 nm in size had a uniform shell of 5 nm thickness [16]. Thus, the shell layer of PTX$x$@HAp is considered to be uniformly stacked in the direction of the ordered arrangement of the $c$-axis of the HAp crystal.

In the FT–IR spectra of PTX$x$@HAp and PLA (figure 4$a$), FT-IR bands corresponding to phosphate groups [10,19,20] and PLA [21,22] were observed; these were assigned to O–P–O bending in PO$_4$ (560, 600 cm$^{-1}$), the PO$_4$ symmetric stretching mode (1025 cm$^{-1}$), the C=O bending mode (750 cm$^{-1}$), the

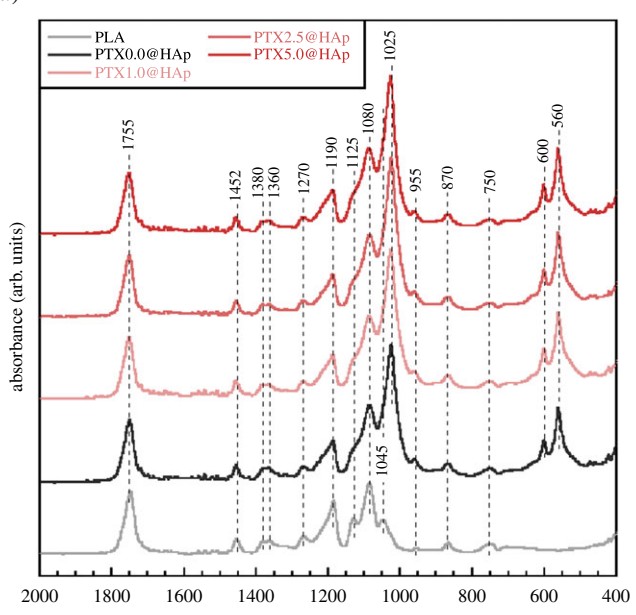

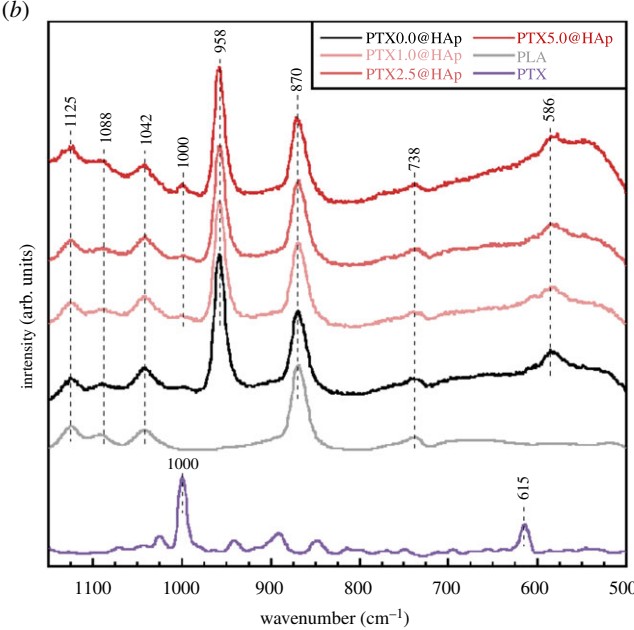

**Figure 4.** (*a*) FT–IR and (*b*) Raman spectra of various PTX*x*@HAp assemblies.

C–COO stretching mode (870 cm$^{-1}$), CH$_3$ rocking with the C–C stretching mode (955 cm$^{-1}$), the C–CH$_3$ stretching mode (1045 cm$^{-1}$), the COC symmetric stretching mode (1080 cm$^{-1}$), the CH$_3$ asymmetric bending mode (1125 cm$^{-1}$), the COC asymmetric stretching mode with CH$_3$ asymmetric bending mode (1190 cm$^{-1}$), the CH bending mode with COC stretching mode (1270 cm$^{-1}$), the CH and CH$_3$ bending mode (1360 cm$^{-1}$), the CH$_3$ symmetric bending mode (1380 cm$^{-1}$), the CH$_3$ asymmetric bending mode (1452 cm$^{-1}$) and the C=O stretching mode (1755 cm$^{-1}$). In the laser Raman spectra of PTX*x*@HAp, PLA and PTX (figure 4*b*), bands corresponding to phosphate groups [23,24] and PLA [21,22] were observed; these were assigned to the symmetric stretching mode of the P–O bond (586 cm$^{-1}$), the PO$_4$ symmetric stretching mode (958 cm$^{-1}$), the C=O bending mode (738 cm$^{-1}$), the C–COO stretching mode (870 cm$^{-1}$), the C–CH$_3$ stretching mode (1042 cm$^{-1}$), the C–O–C symmetric stretching mode (1088 cm$^{-1}$) and the CH$_3$ rocking mode (1125 cm$^{-1}$). The FT–IR and Raman spectra of PTX*x*@HAp showed bands corresponding to the phosphate group, PLA and PTX. The peaks for the phosphate groups observed from the FT–IR (560, 600, 1025 cm$^{-1}$) and Raman (586, 958 cm$^{-1}$) spectra

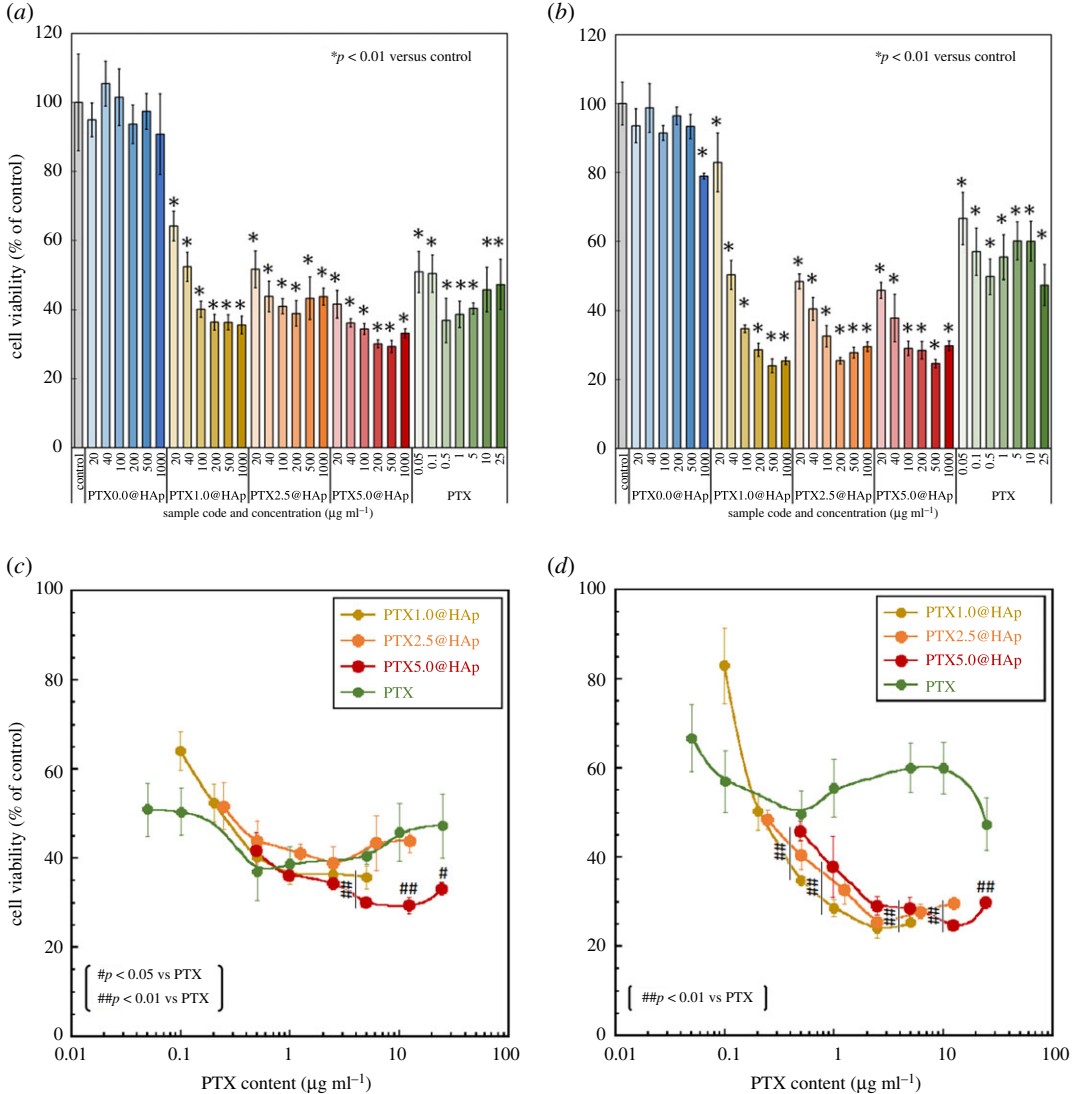

**Figure 5.** Viability of 4T1 cells cultured with PTX$x$@HAp or PTX-containing medium for (*a*) 24 h and (*b*) 48 h, and their viability at (*c*) 24 h and (*d*) 48 h as a function of the PTX content. Error bars represent the standard deviation. Student's *t*-test was used for comparisons between (*a*,*b*) PTX$x$@HAp and the control ($n = 5$, *$p < 0.01$), and (*c*,*d*) between PTX$x$@HAp and PTX solutions of similar concentration ($n = 5$, #$p < 0.05$, ##$p < 0.01$).

in PTX$x$@HAP corresponded to the tetrahedral arrangement of $PO_4^{3-}$, which is the representative phosphate structure in HAp. The intensity of the bands corresponding to PLA in PTX$x$@HAp was not significantly different from that of free PLA. Additionally, the band positions corresponding to PLA in PTX$x$@HAp were not shifted in relation to the band position of free PLA. Therefore, we had successfully constructed PTX$x$@HAp core–shell nanoparticles composed of a PLA core and an HAp shell.

PTX$x$@HAp with $x \geq 1$ showed bands corresponding to $sp^3$-hybridized C–C vibration (1000 cm$^{-1}$) [25], which is the main band for PTX. The intensity of 1000 cm$^{-1}$ increased with increasing PTX content, indicating that PTX had been successfully incorporated into PTX$x$@HAp. Moreover, the amount of anti-cancer drug loaded can be manipulated through the introduction amount. The PLA/HAp particles reported previously could load 750 wt% of a hydrophobic model drug [9]; the PTX$x$@HAp assembly can facilitate a higher PTX-loading amount. Consequently, PTX$x$@HAp core–shell nanoparticles were successfully prepared without morphological changes caused by the incorporation of PTX, and the nanoparticles are expected to accumulate in tumour tissue via the EPR effect.

The viability of 4T1 cells cultured with the PTX$x$@HAp or free PTX solutions is shown in figure 5*a*,*b*. There were no significant differences in cell viability percentages between the PTX0.0@HAp-exposed cells and control cells. By contrast, cells exposed to PTX$x$@HAp ($x \geq 1$) exhibited significantly lower viability percentages compared with the control cells. This indicated that the PTX in PTX$x$@HAp

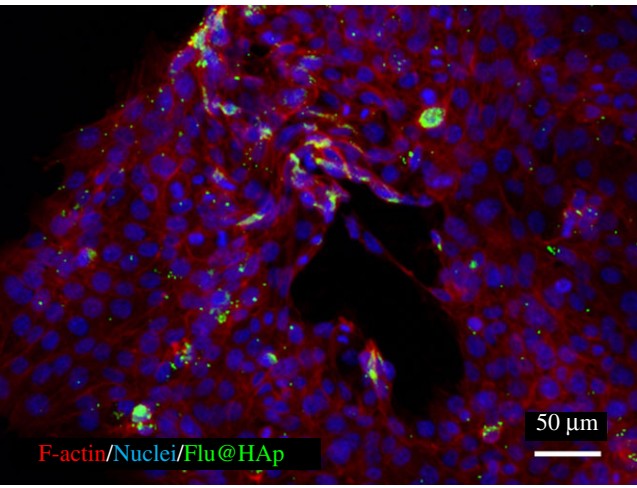

F-actin/Nuclei/Flu@HAp

50 μm

**Figure 6.** Fluorescence image of 4T1 cells cultured with Flu@HAp containing medium. Red, F-actin; blue, nuclei; green, Flu@HAp core–shell nanoparticle (fluorescein).

($x \geq 1$) had been released from the nanoparticles and executed its cytotoxic activity against the cancer cells. This was due to the high solubility of the HAp shell in the acidic environment of the 4T1 cells [5,6,11].

Figure 5c,d shows the viability of cells cultured with the PTX$x$@HAp ($x \geq 1$) and PTX solutions as a function of PTX content. The PTX content of PTX$x$@HAp was calculated using the HAp : (PLA + PTX) weight ratio of 1 : 1, as measured by TG analysis, where the PTX content represents the full amount loaded onto the particles. The percentage viability of cells exposed to PTX$x$@HAp ($x \geq 1$) at 24 h was similar to or slightly lower than that of cells exposed to a similar concentration of PTX solution. In particular, cells cultured with PTX5.0@HAp with a PTX content greater than 5 μg ml$^{-1}$ exhibited significantly lower viability at 24 h compared with those exposed to PTX solutions of a similar concentration. At 48 h, the cells cultured with PTX$x$@HAp ($x \geq 1$) with a PTX content greater than 0.5 μg ml$^{-1}$ also exhibited significantly lower viability than the cells exposed to equivalent PTX solutions. Cell viability under PTX$x$@HAp ($x \geq 1$) exposure generally decreased with increasing culture time (exception of the PTX1.0@HAp with 20 μg ml$^{-1}$ PTX group), whereas that under PTX solution exposure increased with time. The cytotoxic activity of PTX$x$@HAp ($x \geq 1$, PTX content > 0.5 μg ml$^{-1}$) against 4T1 cells lasted up to 48 h (as cell viability decreased from 24 to 48 h), whereas the viability of cells exposed to the PTX solution increased from 24 to 48 h. Figure 6 shows fluorescence images of 4T1 cells cultured with Flu@HAp-containing medium. The core–shell nanoparticles were attached and/or incorporated into the cells. PTX$x$@HAp nanoparticles can be located around and/or inside of 4T1 cells, similar to Flu@HAp. As a result, PTX concentration around the cells exposed to PTX$x$@HAp ($x \geq 1$) may be larger than those around the cells exposed to PTX solutions. Additionally, the core–shell nanoparticles showed sustained releasability of loaded drug [9]. Therefore, the PTX$x$@HAp ($x \geq 1$) nanoparticles exhibited cytotoxic activity to cancer cells until 48 h, due to sustain releasability of PTX.

Luo *et al.* [26] reported that the viability of 4T1 cells cultured with epigallocatechin gallate in combination with PTX (0.9 μg ml$^{-1}$) was only 20%, and their system also successfully decreased the tumour volume and weight *in vivo*. Our PTX$x$@HAp (PTX content > 1.0 μg ml$^{-1}$) construct resulted in a cell viability percentage of approximately 20% at 48 h, indicating its effectiveness for inhibiting tumour cell growth. These results suggest that these PTX$x$@HAp core–shell nanoparticles can be carrier candidates for various water-insoluble drugs.

# 4. Conclusion

PTX-containing PLA/HAp core–shell nanoparticles were prepared using a surfactant-free method. The approximately 80 nm size of the PTX$x$@HAp assembly could be expected to facilitate passive targeting of the drug to tumour tissues via the EPR effect. According to the structural analysis, PTX$x$@HAp consists of a PLA core and an HAp shell, and the amount of PTX loaded onto the nanoparticles can be manipulated through the amount introduced in the fabrication process. PTX$x$@HAp exhibited cytotoxic activity against cancer cells. This was due to the pH sensitivity of the HAp shell, which

dissolves in the typically acidic environment of cancer cells, thereby releasing the drug. The cytotoxic activity of PTX$x$@HAp ($x \geq 1$) against 4T1 cells could be sustained for 48 h and is, therefore, expected to be suitable for use in the inhibition of tumour cell growth.

Data accessibility. The datasets supporting this article have been uploading as part of the electronic supplementary material.

Authors' contributions. S.L. carried out the nanoparticle synthesis and structural analysis, and cell culture test, participated in the design of the study and drafted the manuscript; T.M., A.S.-N. and K.K. participated in the design of the study and critically revised the manuscript. F.N. conceived of the study, designed the study, coordinated the study and helped draft the manuscript. All authors gave final approval for publication and agree to be held accountable for the work performed therein.

Competing interests. The authors have no conflict of interests to declare.

Funding. This work was supported by JST A-STEP grant no. JPMJTS1624.

Acknowledgements. The authors acknowledge Mr T. Matsubara (Nagoya Institute Technology) for providing TEM analysis of the nanoparticles.

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
