## [Peer Review File · Royal Society Open Science]

Review History

RSOS-202030.R0 (Original submission)

Review form: Reviewer 1

Is the manuscript scientifically sound in its present form?

Yes

Are the interpretations and conclusions justified by the results?

Yes

Is the language acceptable?

Yes

Do you have any ethical concerns with this paper?

No

Have you any concerns about statistical analyses in this paper?

No

Recommendation?

Accept with minor revision (please list in comments)

Comments to the Author(s)

The work reports an original approach to the synthesis of new biodegradable core-shell NPs of PTX_x/PLA@HAp as carriers for water-insoluble drugs. The topic of the paper is relevant, and the main results are sufficiently novel to recommend this work for publication. However, some remarks and comments must be considered before accepting.

1. Are there any experimental results that directly confirm the elemental composition of the obtained LPA@HAp NPs?
2. The size distribution of synthesized NPs should be added to the SEM results (Fig. 1) in the histogram form or any other.
3. Does the broadening of the XRD lines confirm the SEM results? Can you estimate the characteristic crystallite size of HAp? In other words, is HAp as a part of composite LPA@HAp NPs monocrystalline or polycrystalline?
4. According to the presented results, the relative spatial arrangement of the LPA and HAp components of composite NPs remains unclear. TEM images could confirm the core-shell structure of nanoparticles claimed by the authors, as well as estimate the characteristic thickness of the shell. See, for example, doi:10.1016/j.jallcom.2020.157812.
5. Do you suppose the possibility of "texturing" NPs along the c axis? The SEM results show a partial alignment of some NPs in a row. If not, then a more detailed explanation of the nonuniform broadening of some XRD lines is required, which is usually associated with the anisotropic shape of nanocrystals. See, for example, doi:10.1016/j.ceramint.2020.06.233.
6. The designation of some results with * (Fig. 4) should be explained in the figure caption.
7. What is the reason for the inhomogeneous distribution of Flu@HAp core-shell NPs over the cell culture?

After taking into account the comments above, this manuscript can be recommended for publication in the Journal. A second review is optional, but this is at the discretion of the Editor.

Review form: Reviewer 2**Is the manuscript scientifically sound in its present form?**

Yes

Are the interpretations and conclusions justified by the results?

Yes

Is the language acceptable?

Yes

Do you have any ethical concerns with this paper?

Yes

Have you any concerns about statistical analyses in this paper?

No

Recommendation?

Accept as is

Comments to the Author(s)

The manuscript entitled Development of paclitaxel-loaded poly(lactic acid)/hydroxyapatite core-shell nanoparticles as a stimuli-responsive drug delivery system has been very describe very well. Authors carried out a fundamental investigation on the in vitro properties of PLA/HAp core-shell nanoparticles loaded with an anticancer agent. Authors observed the cytotoxic activity of PTX \times @HAp ($x \geq 1$) against 4T1 cells could be sustained for 48 h and is therefore expected to be suitable for use in the inhibition of tumor cell growth. All the data is supportive for this study. I would like to recommend the manuscript can be accepted in current form.

Decision letter (RSOS-202030.R0)

Dear Dr Lee:

Title: Development of paclitaxel-loaded poly(lactic acid)/hydroxyapatite core-shell nanoparticles as a stimuli-responsive drug delivery system
Manuscript ID: RSOS-202030

Thank you for submitting the above manuscript to Royal Society Open Science. On behalf of the Editors and the Royal Society of Chemistry, I am pleased to inform you that your manuscript will be accepted for publication in Royal Society Open Science subject to minor revision in accordance with the referee suggestions. Please find the reviewers' comments at the end of this email. I apologise that this has taken longer than usual.

The reviewers and handling editors have recommended publication, but also suggest some minor revisions to your manuscript. Therefore, I invite you to respond to the comments and revise your manuscript.

Because the schedule for publication is very tight, it is a condition of publication that you submit the revised version of your manuscript before 19-Feb-2021. Please note that the revision deadline will expire at 00.00am on this date. If you do not think you will be able to meet this date please let me know immediately.

Kind regards,
Dr Laura Smith
Publishing Editor, Journals

On behalf of the Subject Editor Professor Anthony Stace and the Associate Editor Dr Dattatray Late.

RSC Associate Editor:
Comments to the Author:
cept with minor revisions

RSC Subject Editor:
Comments to the Author:
(There are no comments.)

Reviewer comments to Author:

Reviewer: 1

Comments to the Author(s)

The work reports an original approach to the synthesis of new biodegradable core-shell NPs of PTX_x/PLA@HAp as carriers for water-insoluble drugs. The topic of the paper is relevant, and the main results are sufficiently novel to recommend this work for publication. However, some remarks and comments must be considered before accepting.

1. Are there any experimental results that directly confirm the elemental composition of the obtained LPA@HAp NPs?
2. The size distribution of synthesized NPs should be added to the SEM results (Fig. 1) in the histogram form or any other.
3. Does the broadening of the XRD lines confirm the SEM results? Can you estimate the characteristic crystallite size of HAp? In other words, is HAp as a part of composite LPA@HAp NPs monocrystalline or polycrystalline?
4. According to the presented results, the relative spatial arrangement of the LPA and HAp components of composite NPs remains unclear. TEM images could confirm the core-shell structure of nanoparticles claimed by the authors, as well as estimate the characteristic thickness of the shell. See, for example, doi:10.1016/j.jallcom.2020.157812.
5. Do you suppose the possibility of "texturing" NPs along the c axis? The SEM results show a partial alignment of some NPs in a row. If not, then a more detailed explanation of the nonuniform broadening of some XRD lines is required, which is usually associated with the anisotropic shape of nanocrystals. See, for example, doi:10.1016/j.ceramint.2020.06.233.
6. The designation of some results with * (Fig. 4) should be explained in the figure caption.
7. What is the reason for the inhomogeneous distribution of Flu@HAp core-shell NPs over the cell culture?

After taking into account the comments above, this manuscript can be recommended for publication in the Journal. A second review is optional, but this is at the discretion of the Editor.

Reviewer: 2

Comments to the Author(s)

The manuscript entitled Development of paclitaxel-loaded poly(lactic acid)/hydroxyapatite core-shell nanoparticles as a stimuli-responsive drug delivery system has been very describe very well. Authors carried out a fundamental investigation on the in vitro properties of PLA/HAp core-shell nanoparticles loaded with an anticancer agent. Authors observed the cytotoxic activity of

PTX_x@HAp ($x \geq 1$) against 4T1 cells could be sustained for 48 h and is therefore expected to be suitable for use in the inhibition of tumor cell growth. All the data is supportive for this study. I would like to recommend the manuscript can be accepted in current form.

Author's Response to Decision Letter for (RSOS-202030.R0)

See Appendix A.

Decision letter (RSOS-202030.R1)

Dear Dr Lee:

Title: Development of paclitaxel-loaded poly(lactic acid)/hydroxyapatite core-shell nanoparticles as a stimuli-responsive drug delivery system
Manuscript ID: RSOS-202030.R1

It is a pleasure to accept your manuscript in its current form for publication in Royal Society Open Science. The chemistry content of Royal Society Open Science is published in collaboration with the Royal Society of Chemistry.

On behalf of the Subject Editor Professor Anthony Stace and the Associate Editor Dr Dattatray Late.

RSC Associate Editor
Comments to the Author:
Accept as is

Reviewer(s)' Comments to Author:

Appendix A

Feb. 12th, 2021

Reply to the comments by Reviewer,

Thank you for your valuable and careful comments.

We have revised our manuscript, referring your comments.

Revised parts were **yellow-highlighted**.

Comment 1: Are there any experimental results that directly confirm the elemental composition of the obtained PLA@HAp NPs?

Reply: We added elemental composition of PTX0.0@HAp, and the experimental methods and results shown in page 5 line 8-11 and page 8 line 3-4, respectively, as shown below.

Experimental methods

Removed supernatant of PTX0.0@HAp was filtered using ultrafiltration (Vivaspin 20, Sartorius, 10 kDa MWCO). The filtered supernatant was measured using inductively coupled plasma optical emission spectrometer (IRIS Advantage, Nippon Jarrell-Ash) to confirm Ca/P ratio of the particle.

Results

The Ca/P ratio of PTX0.0@HAp was approximately 1.63, which was similar to stoichiometric composition of hydroxyapatite with the value of 1.67.

Comment 2: The size distribution of synthesized NPs should be added to the SEM results (Fig. 1) in the histogram form or any other.

Reply: We added particle size distribution as histogram in Fig. 1 (e-h), as shown below.

Figure 1: SEM images of (a) PTX0.0@HAp, (b) PTX1.0@HAp, (c) PTX2.5@HAp, and (d) PTX5.0@HAp. Particle diameter distribution of (e) PTX0.0@HAp, (f) PTX1.0@HAp, (g) PTX2.5@HAp, and (h) PTX5.0@HAp. Bar graphs represent appearance frequency, and blue solid lines represent cumulative frequency.

Comment 3: Does the broadening of the XRD lines confirm the SEM results? Can you estimate the characteristic crystallite size of HAp? In other words, is HAp as a part of composite PLA@HAp NPs monocrystalline or polycrystalline?

Reply: We revised broadening of XRD, and results and discussion shown in page 8 line 4-10, as shown below. The HAp crystal in PTXx@HAp core-shell nanoparticles has deformation, since HAp shell covered spherical PLA. Thus, crystalline size of HAp is difficult to estimate present XRD results. However, the crystalline analysis of PLA/HAp core-shell nanoparticle is in preparation of submission for paper.

The XRD patterns of PTXx@HAp (Fig. 3) showed good agreement with those previously established for HAp (JCPDS Database File No. 9-432), and also showed similar patterns with our previous PLA/HAp core-shell nanoparticles.⁹ The XRD patterns of PTXx@HAp were not sharp, which similar with biological hydroxyapatite¹⁷, and indicated that HAp shell had low crystallinity. Additionally, the HAp shell formed on end group of PLA⁹, *i.e.* carboxyl group¹⁸, and the shell composed polycrystalline HAp.

Comment 4: According to the presented results, the relative spatial arrangement of the PLA and HAp components of composite NPs remains unclear. TEM images could confirm the core-shell structure of nanoparticles claimed by the authors, as well as estimate the characteristic thickness of the shell. See, for example, doi:10.1016/j.jallcom.2020.157812.

Reply: We added TEM image of PTX0.0@HAp in Fig. 2, and the experimental methods and results shown in page 5 line 18-20 and page 7 line 14-17, respectively, as shown below.

Experimental methods

PTX0.0@HAp solution directly dropped on carbon grid and dried, subsequently the sample was observed with transmission electron microscope (TEM, JEM-2100F, JEOL).

Results

TEM image of PTX0.0@HAp is shown in Fig. 2. Gray spherical parts, *i.e.* PLA core, were approximately 35 nm diameter, and black shell parts, *i.e.* HAp shell, were approximate 6 nm thickness. These sizes were showed good agreement with our previous reports, where the PLA/HAp core-shell nanoparticles of 50 nm in diameter had a shell of 5 nm thickness.¹⁶

Comment 5: Do you suppose the possibility of "texturing" NPs along the c axis? The SEM results show a partial alignment of some NPs in a row. If not, then a more detailed explanation of the nonuniform broadening of some XRD lines is required, which is usually associated with the anisotropic shape of nanocrystals. See, for example, doi:10.1016/j.ceramint.2020.06.233.

Reply: The broadening XRD patterns for HAp is observed in biological hydroxyapatite, as discussed Comment 3. The particle aggregation is occurred during drying process, when preparation SEM samples. Thus, the partial alignment of PTXx@HAp in SEM images is not affect to c-axis arrangement. We tried to analysis diffraction patterns from TEM, however, the particle was

destroyed due to very weak for heat and/or electron beam when analysis present particle. We are under preparation of the papers for the crystalline analysis of PLA/HAp core-shell nanoparticle by TEM analysis with modifying analysis process.

Comment 6: The designation of some results with * (Fig. 4) should be explained in the figure caption.

Reply: We revised figure caption, page 14 line 9-14, as shown below.

Figure 5: Viability of 4T1 cells cultured with PTXx@HAp or PTX containing medium for (a) 24 h and (b) 48 h, and their viability at (c) 24 h and (d) 48 h as a function of the PTX content. Error bars represent the standard deviation. Student's t-test was used for comparisons between (a, b) PTXx@HAp and the control ($n = 5, * : p < 0.01$), and (c, d) between PTXx@HAp and PTX solutions of similar concentration ($n = 5, \# : p < 0.05, \#\# : p < 0.01$).

Comment 7: What is the reason for the inhomogeneous distribution of Flu@HAp core-shell NPs over the cell culture?

Reply: The particles may remove during washing process. Because, the particle deposited during cell cultivation, and necessary to wash strongly to remove them.

We hope this revised manuscript will be satisfied for accepting.

Your truly,
Sungho Lee